# Oil Extracted from Spent Coffee Grounds as a Green Corrosion Inhibitor for Copper in a 3 wt% NaCl Solution

**Ghada Rouin [1], Makki Abdelmouleh [2], Abdulrahman Mallah [3] and Mohamed Masmoudi [1,4,*]**

1   Laboratory of Electrochemistry and Environment (LEE), National Engineering School, BPW, University of Sfax, Sfax 3038, Tunisia; ghada.rouin2016@gmail.com
2   Laboratory of Materials Sciences and Environment (LMSE), Faculty of Sciences of Sfax, University of Sfax, Sfax 3038, Tunisia; makki.abdmouleh@yahoo.fr
3   Department of Chemistry, College of Science, Qassim University, P.O. Box 6644, Buraydah Almolaydah, Buraydah 51452, Saudi Arabia; a.mallah@qu.edu.sa
4   Preparatory Institute for Engineering Studies of Sfax, BP 805, University of Sfax, Sfax 3018, Tunisia
*   Correspondence: med_mmasmoudi@yahoo.fr

**Abstract:** The aim of this research is to study the use of the spent coffee grounds (SCGs) as a novel and environmentally friendly corrosion inhibitor. The oily fraction obtained from decoction extraction was subjected to analysis using Fourier transform infrared spectroscopy (FTIR) and thermogravimetric analysis (TGA). The inhibitive action of SCG extract on the corrosion of copper in a 3 wt% NaCl solution was studied via potentiodynamic polarization and electrochemical impedance spectroscopy (EIS). The research findings elucidate that the extract derived from SCGs functions as a cathodic inhibitor, primarily impeding the diffusion of oxygen molecules towards the copper substrate. Notably, the inhibitory effectiveness exhibits an improvement with ascending concentrations of the SCG extract. This augmentation culminates in a remarkable 96% inhibition efficiency when the SCG extract concentration reaches 0.6 g/L. Furthermore, it is observed that the SCG extract undergoes adsorption onto the copper surface, a phenomenon that conforms to the Langmuir adsorption isotherm.

**Keywords:** spent coffee grounds; bio-oil; FTIR; voltammetry around OCP; EIS

## 1. Introduction

Copper is extensively used in various industries, particularly in electronics and integrated circuit production, owing to its exceptional electrical and thermal conductivity properties [1]. Nevertheless, copper's susceptibility to corrosion, especially in the presence of aggressive ions like chloride, imposes substantial limitations on its application [2].The preservation of metals against corrosion ranks among the foremost imperatives in the industrial sector. This persistent challenge retains its pertinence in a wide range of industrial applications and products, bearing consequences in terms of component and system degradation, which can ultimately lead to operational failures. As corrosion poses a risk to economic, environmental, and safety issues in a variety of industrial applications, it requires quick action.

The investigation of copper corrosion in aqueous media is primarily aimed at serving industrial corrosion control measures [3–5]. Organic compounds containing atoms of sulfur, phosphorus, nitrogen, and oxygen constitute a significant proportion of the inhibitors employed. Their inhibitory impact is frequently ascribed to their adsorption onto the metal surface [6]. Present research endeavors are increasingly focused on exploring the potential of green inhibitors. These inhibitors possess non-toxic and biodegradable properties, rendering them favorable as environmentally friendly corrosion inhibitors. A promising alternative to chemical-based compounds lies in the utilization of wasted food as an

ecologically sustainable resource. Consequently, efforts are underway to develop corrosion protection methods that adhere to the principles of "green chemistry" [7–12].

Coffee, being the second most significant commercial commodity, generates a substantial volume of spent coffee grounds (SCGs) annually [13,14]. Extensive research in the literature highlights the considerable potential of used coffee grounds in producing value-added energy and non-energy-related products due to their significant content of fatty acids, cellulose, esters, lignin, and hemicelluloses [15–17]. Diverse investigations have been conducted to examine the corrosion-inhibitory efficacy of SCG extract across a spectrum of metals and within various environmental conditions [18–22]. According to research by Bouhlel et al. [18,23], the hydro-alcoholic extract of used coffee grounds effectively protects C38 steel against corrosion in 1 M hydro-alcoholic acid. This effect may be attributed to the extraction process and the solvent's polarity.

Notably, one study focused on copper as the substrate and NaCl as the electrolyte. Velazquez-Torres et al. [21] utilized a fatty amide originating from coffee oil as a corrosion inhibitor, synthesized through the direct aminolysis of oil extract and hydroxyethyl ethylenediamine. Their findings demonstrated that the N-[2-[(2-hydroxyethyl) amino] ethyl]-amide derived from coffee residue functions as a mixed inhibitor for copper in a 3.5 wt% NaCl solution. The novelty in our research lies in the direct utilization of SCG extract oil as a corrosion inhibitor for copper in chloride-rich media, circumventing the need for aminolysis as typically required. The methodology employed in our study not only obviates the necessity for additional specialized equipment but also facilitates the acquisition of SCG oil through a straightforward extraction procedure.

The primary objective of this work is to investigate the corrosion inhibition of copper in a saline solution using the oil extract derived from spent coffee grounds (SCGs) at various inhibitor concentrations. To achieve this, several electrochemical techniques, including potentiodynamic polarization over a wide potential range, voltammetry around the open-circuit potential (OCP), and electrochemical impedance spectroscopy (EIS), were employed.

## 2. Experimental

### 2.1. Materials

Spent coffee grounds (virgin SCGs) were collected after the infusion of roasted coffee beans (Boundin, Tunisia) in an automatic coffee machine. To ensure the removal of impurities, a pretreatment procedure was implemented. This involved washing the virgin SCGs with lukewarm water, followed by a drying phase in an oven maintained at 60 °C for 24 h. Subsequently, an extraction method was employed, entailing a three-hour decoction [20] in 100 mL of n-hexane, maintained at 78 °C. The resultant extract was then subjected to filtration, resulting in what is referred to as "de-oiled SCG". The hexane extract, in turn, underwent further processing through evaporation using a vacuum-assisted rotavaporator (SCG oil), as illustrated in Figure 1 (step 1). To prepare the inhibitor solution, a specific mass of the obtained oil was dissolved in tween 80 (m, 1/2 m), resulting in concentrations of 0.2 g/L, 0.4 g/L, and 0.6 g/L. The corrosive solution (3 wt% NaCl) was made using analytical-grade sodium chloride in distilled water, treated as a blank for comparison. The solution tests are freshly prepared before each experiment through adding the inhibitor solutions to the corrosive media. The influence of the SCG oil extract concentration was studied at room temperature (25 ± 2 °C), as depicted in Figure 1 (step 2).

### 2.2. Characterization of the SCG Extract

Utilizing a Perkin Elmer spectrometer, Fourier transform infrared spectroscopy (FTIR) analysis was conducted. Each individual sample underwent scanning across a wavenumber spectrum spanning from 400 $cm^{-1}$ to 4000 $cm^{-1}$, employing a resolution of 2 $cm^{-1}$ per spectrum. Thermogravimetric analysis (TGA) curves of the SCG samples were obtained using a TA Instruments TGA Q500 from Perkin Elmer (Waltham, MA, USA). Analysis of the SCG samples encompassed a temperature interval spanning from 30 to 800 °C, employing a controlled heating rate of 10 °C/min, all conducted under an ambient air atmosphere.

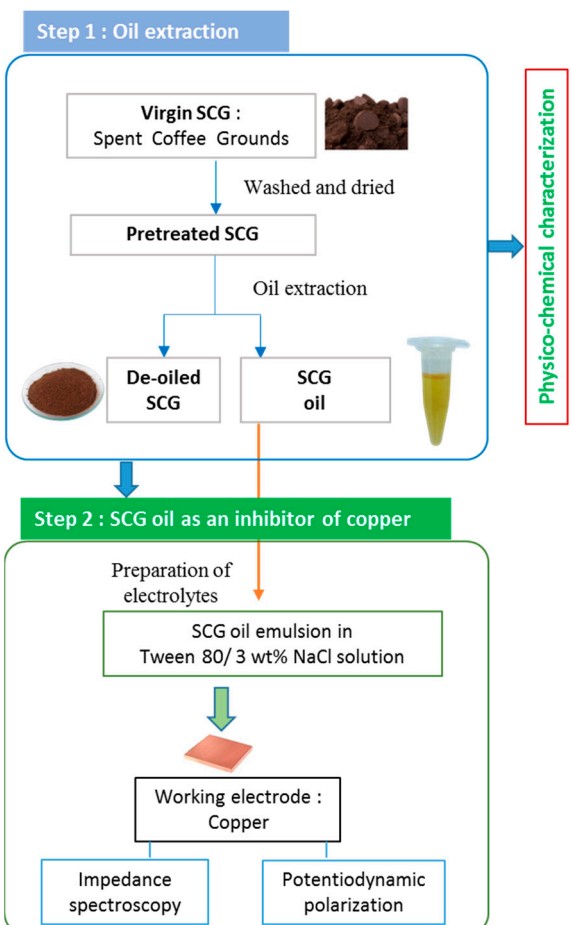

**Figure 1.** The two-step process of SCG oil extraction and its usage as a corrosion inhibitor for copper.

*2.3. Electrochemical Measurements*

Electrochemical measurements were conducted using an electronic potentiostat galvanostat, specifically the Radiometer PST050 (Radiometer, Copenhagen, Danmark), which was controlled through Volta Lab software (Voltamaster 4). The electrochemical analyses were performed under computer control to ensure precise and accurate measurements.

The electrochemical techniques (potentiodynamic polarization and electrochemical impedance spectroscopy) were carried out in a conventional three-electrode cell. Pure copper (99.99%), platinum foil, and a saturated calomel electrode (SCE) were employed as the working, counter, and reference electrodes, respectively. The working surface of the copper electrodes, with an area of 100 mm², was carefully masked with epoxy resin to expose only the desired area.

Prior to each electrochemical test, the exposed copper surface was carefully prepared. This involved polishing the surface using SiC paper, starting from grade 240 and gradually progressing to grade 1200. The surface was then thoroughly rinsed with distilled water, followed by degreasing with acetone. Finally, the electrodes were dried using warm air.

The potentiodynamic polarization curves were plotted from −400 to 400 mV/SCE with a scan rate of 0.5 mv/s. To ensure the stability of the working electrodes, a 90 min immersion period at the open circuit potential (OCP) was implemented prior to all experiments.

The selection of the 90 min immersion period was based on a preliminary investigation involving five different immersion times (30, 60, 90, 120, and 150 min) to provide optimal stability and consistent results.

Voltammetry measurements around OCP were conducted after the 90 min immersion of the working electrode at room temperature, maintaining the OCP conditions. The potential was swept from OCP − 60 mV to OCP + 60 mV at a scan rate of dE/dt = 0.5 mV/s.

The polarization curves, used in the limited potential range around OCP, created only a slight disturbance to the metal surface. The experimental curves fit indicated that the anodic and cathodic reactions comply with Tafel's law. The mathematical modeling of the potentiodynamic polarization curve ($\Delta E = \pm 60$ mV) was performed using Equation (1) [24,25]:

$$j = j_\mathrm{a} + j_\mathrm{c} = j_\mathrm{corr}\left[e^{\beta_\mathrm{a}(E-E_\mathrm{corr})} - e^{\beta_c(E-E_\mathrm{corr})}\right] \tag{1}$$

where $\beta_a$ and $\beta_\mathrm{c}$ are the anodic and cathodic Tafel coefficients, in mV/dec, respectively.

For the electrochemical impedance spectroscopy (EIS) measurements, the samples were allowed to stabilize for a 90 min immersion period at the OCP. A sinusoidal signal with an amplitude of 10 mV was applied over a frequency range spanning from 100 kHz to 0.1 Hz, with a 10 mV AC perturbation signal as the excitation.

To obtain the impedance spectra fitting, electrical equivalent circuits were employed. The fitting process was carried out by means of the "Randomize + Simplex" fitting mode, utilizing EC-Lab V10.32 software. The maximum number of iterations was set at 10,000. The assessment of fitting quality relied upon the calculation of chi-square ($\chi^2$) values, which represent the summation of squared differences between experimental and theoretical values.

## 3. Results and Discussion

### 3.1. FTIR Spectral Analysis

The FTIR spectra of different SCG samples, including virgin SCGs, pretreated SCGs, de-oiled SCGs, and SCG oil, are presented in Figure 2. The identified functional groups are listed in Table 1.

Examination of the FTIR spectra of all SCG-based samples has revealed a wide vibration around 3320 cm$^{-1}$, which can be attributed to the stretching vibrations of hydroxyl groups (OH). This indicates the presence of phenols or alcohols stemming from the lignin content of the biomass feedstock [26].

The absorption peaks observed between 2858 cm$^{-1}$ and 2924 cm$^{-1}$ correspond to the asymmetric and symmetric stretching vibrations of the C-H bonds in the aliphatic CH$_2$ groups, confirming the presence of aliphatic compounds in the SCG samples [27].

Furthermore, the FTIR spectrum of the SCG oil exhibited characteristic peaks associated with lipid compounds [27]:

(i)    The ester carbonyl group's vibration (C=O) was observed between 1740 cm$^{-1}$ and 1744 cm$^{-1}$, which is a characteristic feature of carbonyl groups found in lipids, esters, and carboxylic acids [28].

(ii)   The region around 1639 cm$^{-1}$ in the FTIR spectrum corresponds to the presence of C=C bonds.

The vibration bands observed between 1000 and 1100 cm$^{-1}$ correspond to the C-O and C-C-O bonds, which can be attributed to the presence of cellulose, hemicellulose, and lignin in the SCG samples [29]. In the wavenumber range of 1400–900 cm$^{-1}$, various vibrations associated with different bond types, such as C-H, C-O, and C-N, were observed. These vibrations are resulted from the absorption bands of carbohydrates and specific compounds present in coffee beans, such as chlorogenic acids [30,31].

The most relevant changes between the spectra of the virgin SCGs and pretreated SCGs, observed in the regions associated with OH stretching (3315 cm$^{-1}$) and carbonyl stretching vibrations (1744 cm$^{-1}$), confirm the removal of water-soluble compounds during the pretreatment step. The absence of distinctive peaks inherent to lipid compounds within the Fourier transform infrared (FTIR) spectrum of the de-oiled sample provided evidence the coffee oil might be completely extracted through the application of the decoction method employing n-hexane as the solvent [27].

Indeed, the FTIR spectrum of the extracted oil from the coffee ground powder, using the process adopted in our work, clearly shows the presence of all the characteristic bands

corresponding to various groups of oily compounds. Specifically, the FTIR spectrum of the SCG oil obtained from decoction extraction reveals the wavenumber at 3009 cm$^{-1}$, which is assigned to the C-H stretching vibration of *cis* double bonds. This indicates the presence of unsaturated fatty acids in the oil [27,32].

The absorption peaks at 2921 cm$^{-1}$ and 2853 cm$^{-1}$ are typically attributed to the C-H stretching vibrations of alkyl groups, indicating the presence of hydrocarbons in the SCG oil [26]. The presence of triglycerides is confirmed by peaks at 1746 cm$^{-1}$ and 1161 cm$^{-1}$, corresponding to the carbonyl group (-C=O stretch) and ester groups (-C-O stretch), respectively [33].

The C-H bending vibrations of alkane groups are represented by peaks at 1376–1463 cm$^{-1}$. The absorption between 1161 cm$^{-1}$ and 1099 cm$^{-1}$ resulted from to C-O stretching vibrations, which correspond to the ester groups present in the SCG oil. Furthermore, the pic at 721 cm$^{-1}$ indicates the presence of (=C-H) bending vibrations and *cis*-disubstituted alkenes [34].

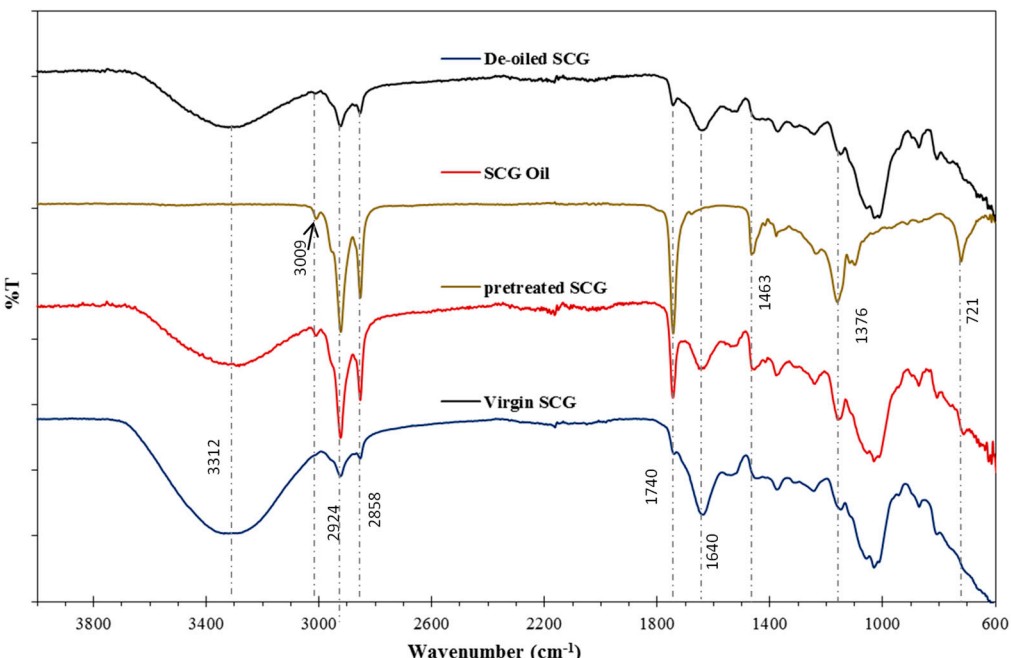

**Figure 2.** FTIR spectra of SCG samples and SCG oil.

**Table 1.** Analytical evaluation of infrared spectra of SCG samples and SCG oil.

| Wavenumbers (cm$^{-1}$) | | Attributions |
|---|---|---|
| SCG Samples | SCG Oil | |
| 3312 | --- | O-H stretching vibration |
| --- | 3009 | C-H stretching symmetric vibration of the *cis* double bonds [35] |
| 2924 | 2921 | Asymmetric and symmetric stretching vibration of C-H bonds of aliphatic CH$_3$ [32,35] |
| 2858 | 2853 | |
| 1740 | 1746 | Stretching vibration of ester carbonyl functional groups (C=O) |
| 1463 | 1463 | Bending vibration of C-H of CH$_2$ and CH$_3$ aliphatic group |
| 1640 | | C=C stretching vibration *cis*-olefins [35] |
| 1369 | 1376 | Bending symmetric vibration of C-H bonds of CH$_2$ groups [35] |
| 1100 | 1161 | Stretching vibration of C-O-C ester groups [36] |
| 1400–900 | 1400–900 | Stretching vibration of (C-O), (C-H), (C-N) [27,30,31] |
| 721 | 721 | =C-H Aliphatic CH$_2$ rocking vibration and *cis* substituted olefin out-of-plane vibration overlapping [32,37] |

### 3.2. Thermogravimetric Analysis

The results of the thermogravimetric analysis (TGA) are presented in Figure 3. The TGA curves for all SCG samples took place mainly in three decomposition stages. The first stage, occurring at temperatures below 130 °C, corresponds to the removal of adsorbed water from the SCG powders. The mass losses during this stage were measured at approximately 47% for virgin SCGs, 5% for pretreated SCGs, and 9% for de-oiled SCGs.

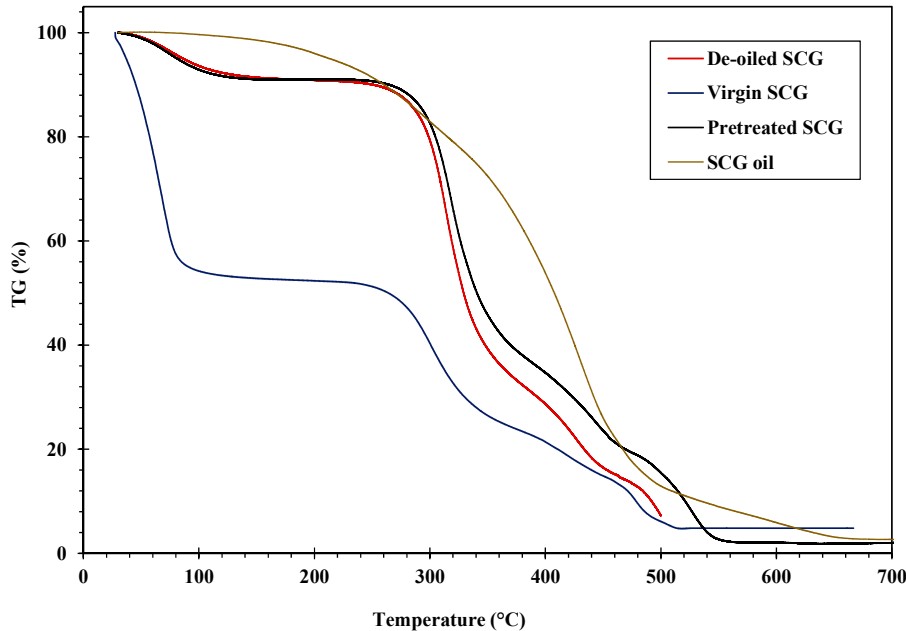

**Figure 3.** Thermogravimetric curves of virgin SCGs, pretreated SCGs, de-oiled SCGs, and SCG oil.

The higher mass loss observed in virgin SCGs can be ascribed to the presence of residual water molecules within the SCG waste structure. The relatively higher mass loss value in de-oiled SCGs compared to pretreated SCGs may be due to the enhanced accessibility of de-oiled SCG powders after the extraction of the oily phase, which created a barrier to the adsorption of water molecules.

The second stage, occurring at around 250 °C, is characterized by the most significant transformation and mass losses. During this stage, the depolymerization and decomposition of polysaccharides (such as cellulose, hemicellulose, and lignin) [38], as well as the decomposition of some oils present in the sample, took place [38].

The third stage, observed at temperatures above 550 °C with a minimal rate of mass loss, indicates the formation of carbonaceous solids [28]. The residual mass obtained above 550 °C is much clearer in the virgin SCG sample, suggesting the presence of solid impurities or inorganic compounds in the SCG waste.

The TGA curve of the SCG oil, analyzed under an air atmosphere (Figure 3), demonstrated no mass loss below 150 °C, indicating the absence of solvents in the extracted oil. The SCG oil's thermal decomposition occurs in two distinct stages between 150 and 650 °C [27,39].

### 3.3. Electrochemical Studies

3.3.1. Potentiodynamic Polarization in the Range of −0.4 to +0.4 V/SCE

Figure 4 demonstrates the potentiodynamic polarization curves of copper electrodes immersed in a 3 wt% NaCl solution for 90 min, both in the absence (blank) and presence of SCG oil extracts at a temperature of 25 °C.

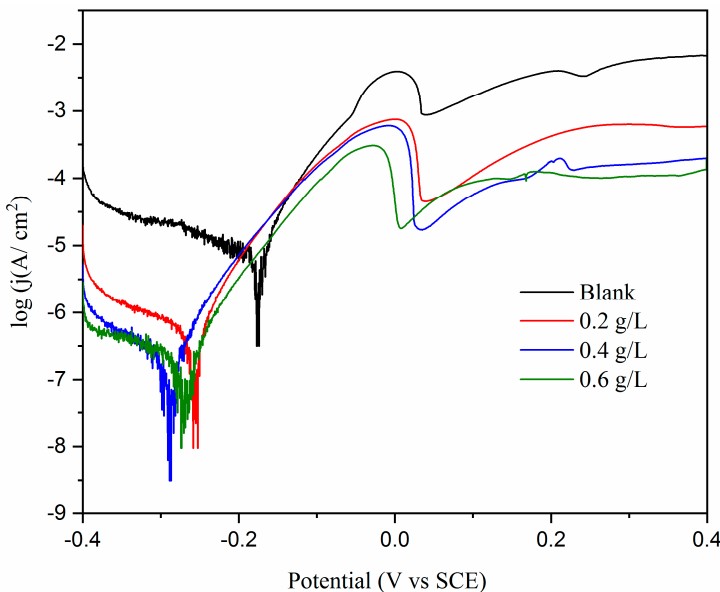

**Figure 4.** Copper's polarization curves exposed to 3 wt% NaCl solutions both in the absence (blank) and presence of different concentrations of SCG extract at a temperature of 25 °C.

Upon examination of this figure, it becomes evident that the cathodic branch of copper in the blank chloride solution can be segmented into two distinct regions. Firstly, region I coincides with the weak polarization region near OCP. Subsequently, region II is characterized by a nearly flat current density, which is imputed to the reduction of dissolved oxygen according to Equation (2).

$$O_2 + 2H_2O + 4e^- \rightarrow 4OH^- \tag{2}$$

Both the electro-dissolution and the soluble diffusion of the metal in the blank solution contribute to the control of the anodic reaction. Consistent with the findings of our earlier study [40], the dissolution process can be appropriately elucidated through the subsequent reactions:

$$Cu \rightarrow Cu^+ + e^- \tag{3}$$

$$Cu^+ + Cl^- \rightarrow CuCl_{(film)} \tag{4}$$

$$CuCl + Cl^- \rightarrow CuCl_{2(surface)}^- \tag{5}$$

Figure 4 demonstrates a notable shift in the cathodic branch of the polarization curves towards more negative potentials and lower current densities in the presence of SCG extract. This shift confirms that the oil extract primarily affects the cathodic reduction of oxygen rather than the oxidation reaction of copper. The measured $E_{corr}$ values for the blank solution, as well as the solutions containing SCG extract concentrations of 0.2 g/L, 0.4 g/L, and 0.6 g/L, are recorded as −176 mV/SCE, −272 mV/SCE, −287 mV/SCE, and −274 mV/SCE, respectively. Consequently, the $E_{corr}$ shift reaches 111 mV/SCE for the 0.4 g/L concentration, which is a significant value that classifies the SCG extract as a cathodic inhibitor [4,24].

### 3.3.2. Voltammetry around OCP (ΔE = ±60 mV)

To better explain the corrosion resistance performance of SCG oil extract, potentiodynamic polarization curves were obtained through immersing the samples in a 3 wt% NaCl solution supplemented with varying concentrations of the oil extract. The polarization curves were obtained within a limited potential range (ΔE = ±60 mV) and were computer-

fitted using the EC-Lab program V10.32 (Bio-Logic), as described in Section 2.3. The fitting process aimed to first determine the corrosion current density ($J_{corr}$), the Tafel coefficients ($\beta_a$ and $\beta_c$), and the corrosion rate (CR) and to later calculate the polarization resistance ($R_p$) and the inhibition efficiency ($\eta$ %) using Equations (6), (7) and (8), respectively.

$$R_p = \frac{B}{J_{corr}} \tag{6}$$

Here, B is a constant that is calculated using the Stern–Geary Equation [41],

$$B = \frac{\beta_c \, \beta_a}{2.303 \, (\beta_c + \beta_a)} \tag{7}$$

$$\eta \, (\%) = \frac{CR^0 - CR}{CR^0} \times 100 \tag{8}$$

where $CR^0$ and CR represent the corrosion rate in both the absence and presence of an inhibitor, respectively.

In Table 2, all of these electrochemical parameters are reported.

**Table 2.** Electrochemical kinetic parameters and inhibition efficiency obtained from potentiodynamic polarization curves ($\Delta E = \pm 60$ mV) after a 90 min immersion in a 3 wt% NaCl solution at RT (~25 °C).

| E | $E_{corr}$ (mV/SCE) | $J_{corr}$ μA cm$^{-2}$ | $\beta_a$ (mV/dec) | $-\beta_c$ (mV/dec) | CR mm year$^{-1}$ | Rp (kΩ·cm$^2$) | η (%) |
|---|---|---|---|---|---|---|---|
| Blank | $-176 \pm 2$ | $5.41 \pm 1.2$ | $40 \pm 13$ | $120 \pm 5$ | $0.063 \pm 1.1$ | $2.40 \pm 1.3$ | – |
| 0.2 g/L | $-272 \pm 4$ | $0.745 \pm 0.3$ | $45.4 \pm 2$ | $62.8 \pm 6$ | $0.00868 \pm 0.3$ | $15.35 \pm 2$ | $86.22 \pm 5$ |
| 0.4 g/L | $-287 \pm 3$ | $0.305 \pm 0.3$ | $59.6 \pm 7$ | $60.7 \pm 14$ | $0.00355 \pm 0.3$ | $42.81 \pm 1.4$ | $94.36 \pm 4$ |
| 0.6 g/L | $-274 \pm 4$ | $0.228 \pm 0.2$ | $46.8 \pm 2$ | $95.9 \pm 2$ | $0.00265 \pm 0.2$ | $59.89 \pm 2.8$ | $95.78 \pm 3$ |

Figure 5 presents a comparison between the experimental polarization curve and the Tafel curve simulated using EC-Lab V10.32 software (Bio-Logic, Seyssinet-Pariset, France) for copper in the presence of SCG oil at a concentration of 0.4 g/L in a saline solution.

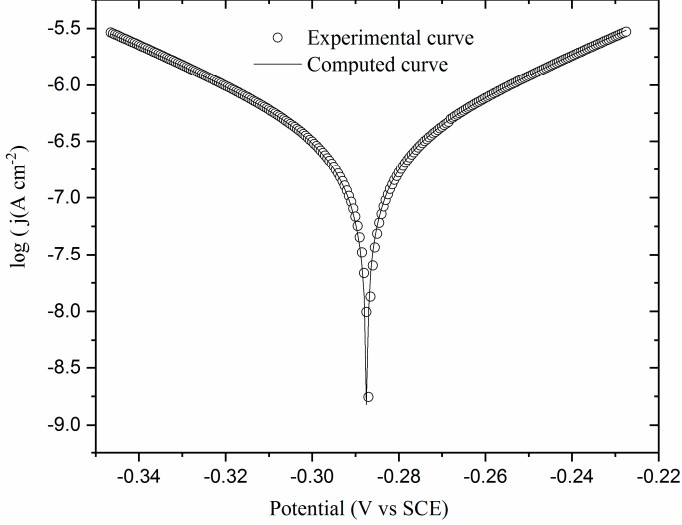

**Figure 5.** Experimental polarization curve around OCP exposed to 3 wt% NaCl solutions containing 0.4 g/L of SCG oil at RT (~25 °C) and computed curve.

In the experimental conditions employed, the cathodic branch of the polarization curves corresponds to oxygen reduction reactions, while the anodic branch represents the dissolution of copper [42]. It is noteworthy that the presence of SCG oil extract has a

minimal impact on the anodic Tafel slopes ($\beta_a$), which remain relatively constant with an average value of $50 \pm 7$ mV/dec.

This finding suggests that the presence of SCG oil does not significantly influence the dissolution of copper. On the other hand, the cathodic Tafel coefficient ($\beta_c$) in the blank solution was determined to be 120 mV/dec. However, upon the addition of the inhibitor, these values decrease to $60.7 \pm 14$ mV/dec for an SCG oil concentration of 0.4 g/L. This decline depicts that the inhibitor primarily affects the cathodic reaction.

The results presented in Table 2 demonstrate a notable decrease in the CR as the inhibitor concentration increases, as indicated by the decrease in the values of $J_{corr}$. The CR reaches a minimum value of 0.0026 mm/year. Moreover, there is a substantial increase in the polarization resistance (Rp), particularly at a concentration of 0.6 g/L, where Rp reaches a value of $59.89 \pm 2.8$ k$\Omega \cdot$cm$^2$.

The inhibition efficiency ($\eta$) also exhibits a remarkable increase, reaching 95.78% at a concentration of 0.6 g/L of SCG oil extract. These findings confirm that SCG oil possesses a strong inhibitory effect on copper corrosion in chloride-containing media. Furthermore, it is evident that higher concentrations of SCG oil extract boost the protective ability against copper corrosion.

### 3.3.3. Electrochemical Impedances Spectroscopy (EIS)

Figure 6 reveals the Nyquist diagrams obtained from electrochemical impedance spectroscopy (EIS) measurements after a 90 min immersion in a 3 wt% NaCl solution, both in the absence and presence of various concentrations of the inhibitor, at a temperature of 25 °C. The Nyquist diagram obtained without the inhibitor (blank) shows a capacitive semicircle at high frequencies and a straight inclined line, which is indicative of a Warburg-type diffusion process, at low frequencies.

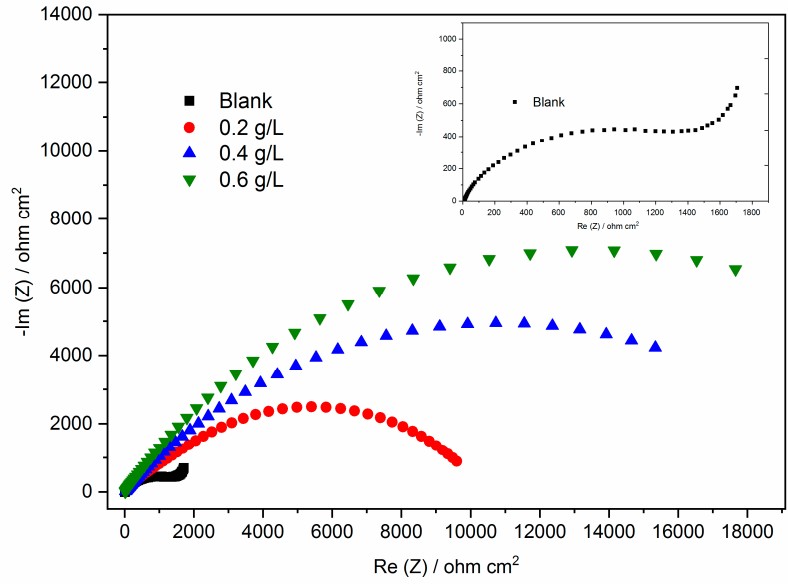

**Figure 6.** Nyquist plots for copper exposed to 3 wt% NaCl solutions both in the absence (blank) and presence of different concentrations of SCG extract at a temperature of 25 °C.

This semicircle represents the combined effects of charge transfer resistance and double-layer capacitance. It should be noted that the capacitive loop is not perfectly circular, which can be imputed to the surface roughness and inhomogeneity of the electrode [24,43,44]. The straight line observed in the low-frequency region can be attributed to the anodic diffusion of soluble copper species (CuCl$_2^-$) from the electrode into the chloride solution, along with the cathodic diffusion of oxygen [25].

The Nyquist curves of copper in the presence of various concentrations of SCG oil extract in a 3 wt% NaCl solution exhibit noticeable distinctions compared to the curve

obtained without SCG oil. This means that the change in the corrosion mechanism occurs upon the addition of the inhibitor. The size of the capacitive loops observed in the Nyquist plots seems to increase with higher concentrations of the extract. In fact, the Nyquist loop diameter for a concentration of 0.6 g/L of SCG oil extract is significantly larger than that observed for other electrode. This observation can be imputed to the formation of a protective layer of the extract on the copper surface, which becomes more compact and adheres firmly as the concentration increases [45]. Consequently, this stable layer serves as a proficient impediment, effectively inhibiting the corrosion of the underlying copper substrate.

Figure 7a depicts the logarithm of impedance amplitude (log $|Z|$) as a function of the logarithm of frequency (log freq). At low frequencies, the impedance amplitude $|Z|$ of bare copper is relatively low (approximately $10^3$ $\Omega$ cm$^2$), indicating its susceptibility to corrosion because the CR and the value of $|Z|$ are inversely related [46]. Conversely, upon the addition of the inhibitor, the $|Z|$ values of copper notably increase, especially at a concentration of 0.6 g/L, reaching approximately $10^4$ $\Omega$ cm$^2$ within the same frequency range. Moreover, at low frequencies, the log $|Z|$ values remain relatively constant. This observation suggests an enhanced resistive response of the copper electrode [23].

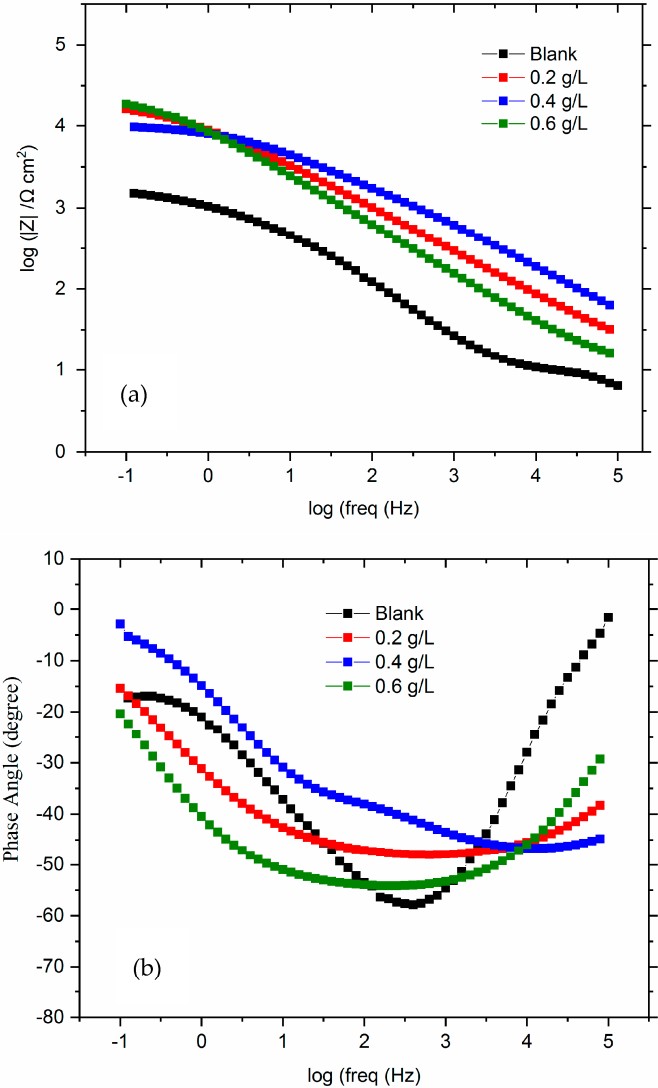

**Figure 7.** Bode plots (**a**) and phase angle (**b**) versus frequency for SCG oil for copper exposed to 3 wt% NaCl solutions at RT (~25 °C).

Figure 7b displays the phase angle plots of copper in the absence and presence of various concentrations of SCG oil extract following a 90 min immersion in 3 wt% NaCl solutions. The phase angle, Arg (Z), representing the phase shift between the current and the potential, is plotted as a function of the logarithm of frequency. In the case of copper immersed in the blank saline solution, the phase angle plot reveals the presence of two distinct time constants. The first time constant, observed in the high-frequency region, corresponds to the relaxation process of the double layer capacitance. The second time constant, observed in the low-frequency region, corresponds to the Warburg diffusion process, which is associated with the corrosion process [47].

The presence of two distinct time constants can be seen on the curves after the addition of the working oil. One time constant is observed at higher frequencies, which can be attributed to the effect of the inhibitor. The other time constant is observed at medium frequencies, associated with the double layer capacity. Consistent with the previous description of the EIS findings, the impedance data were analyzed using two equivalent circuits, as depicted in Figure 8.

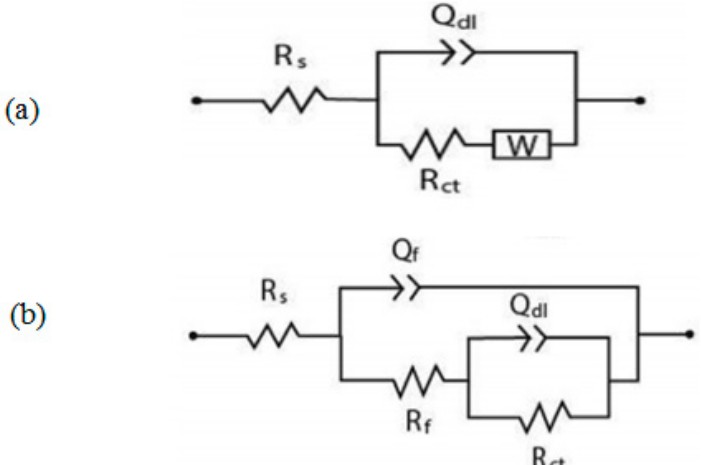

**Figure 8.** Equivalent circuits (**a**) $R_s(Q_{dl}(R_{ct} W))$ (**b**) $R_s(Q_f (R_f (Q_{dl} R_{ct})))$ used to fit the EIS experimental data.

The behavior of copper or copper alloys in chloride-containing solutions has been extensively studied in the literature using different models, both in the absence and presence of inhibitor adsorption [48,49]. In the case of the blank solution, the impedance data of copper can be adequately described using the equivalent circuit shown in Figure 8a. This model includes the solution resistance ($R_s$), charge transfer resistance ($R_{ct}$), double layer capacitance ($Q_{dl}$), and Warburg impedance (W) associated with diffusion processes in the low frequency region [24].

On the other hand, when analyzing the EIS data of copper in solutions containing SCG oil, it was appropriate to use the equivalent circuit $R_s(Q_f(R_f(Q_{dl}R_{ct})))$, as illustrated in Figure 8b. This circuit incorporates the passive film resistance ($R_f$) and passive film capacitance ($Q_f$) components.

The presence of the "dispersion effect" at the solid/liquid interface is evident from the impedance loops observed in Figure 6, where the centers of the loops are located below the real axis. These imperfect semicircles can be attributed to the surface inhomogeneity and roughness of the solid electrode [50]. To accurately fit the EIS data, it is necessary to replace the ideal capacitor with a constant phase element (CPE) that accounts for this non-ideal behavior. The impedance of the CPE can be calculated using Equation (9) [51].

$$Z_{CPE} = \frac{1}{Q_0(j\omega)^n} \tag{9}$$

where $Q_0$ represents the amplitude of the CPE, j symbolizes the imaginary unit, $\omega$ denotes the angular frequency, and the parameter n is assigned to the heterogeneous characteristics of the electrode, stemming from surface topography variations, the creation of porous layers, the adsorption of inhibitors, etc. [52].

The fitted and experimental findings showed excellent agreement ($\chi^2$ values lower than $10^{-3}$). Figure 9 shows an example of the experimental and computer fitting results. The model shows a good agreement with the experimental results for both Nyquist and Bode plots shown in Figure 9a and 9b, respectively.

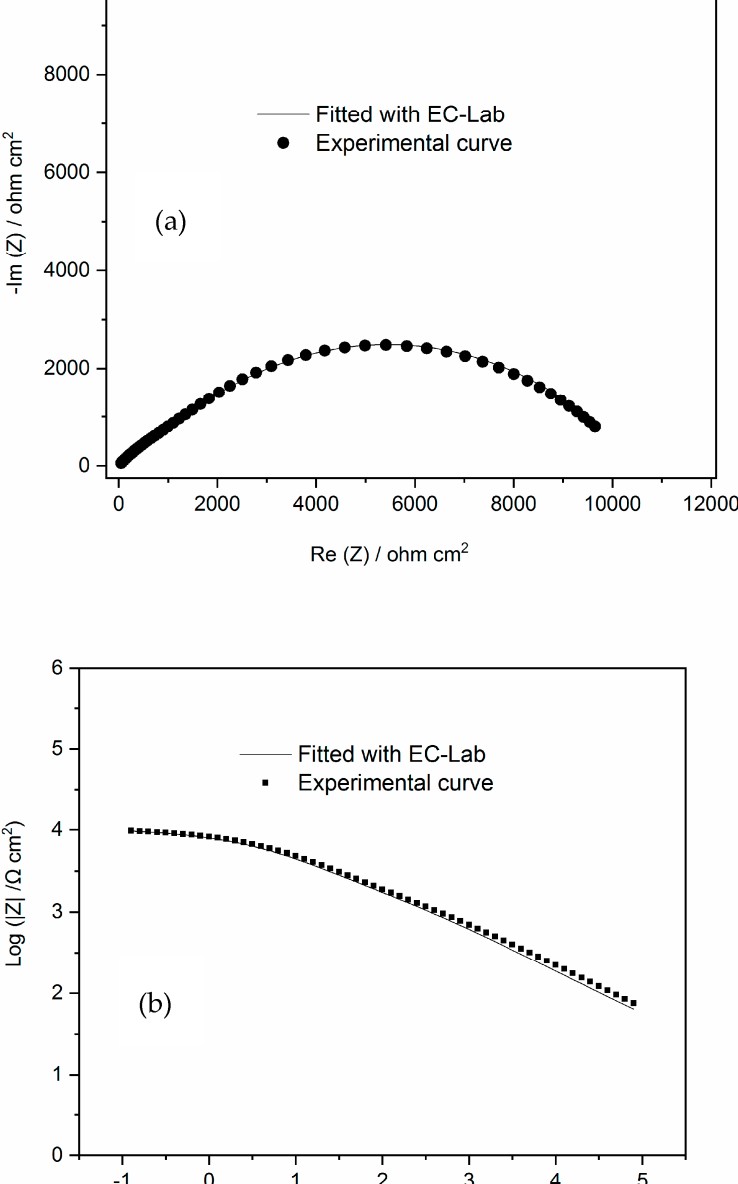

**Figure 9.** Experimental Nyquist (**a**) and Bode plots (**b**) and their mathematical fitting: an example of copper exposed to 3 wt% NaCl solutions with SCG oil concentration of 0.2 g/L.

The electrochemical parameters obtained after computer fitting EIS data through electric circuits shown in Figure 8 are listed in Table 3.

**Table 3.** Electrochemical parameters and inhibition efficiency obtained from EIS study of copper electrode after a 90 min immersion in 3 wt% NaCl solutions both in the absence (blank) and presence of different concentrations of SCG extract at a temperature of 25 °C.

| | Blank | 0.2 g/L | 0.4 g/L | 0.6 g/L |
|---|---|---|---|---|
| $R_s$ ($\Omega$ cm$^2$) | $6.915 \pm 0.3$ | $7.03 \pm 0.8$ | $8.4 \pm 3.6$ | $9.02 \pm 2.2$ |
| $R_{ct}$ ($\Omega$ cm$^2$) | $1466 \pm 80$ | $6725 \pm 200$ | $13{,}985 \pm 580$ | $17{,}980 \pm 630$ |
| $Q_{dl} \times 10^{-6}$ (F cm$^{-2}$ s$_{dl}^n$) | $145 \pm 20$ | $7.42 \pm 2.8$ | $6.836 \pm 4.1$ | $7.2 \pm 3.5$ |
| $n_{dl}$ | $0.687 \pm 0.4$ | $0.67 \pm 0.02$ | $0.4 \pm 0.03$ | $0.53 \pm 0.06$ |
| $R_f$ ($\Omega$ cm$^2$) | - | $3698 \pm 7$ | $7821 \pm 28$ | $8932 \pm 37$ |
| $Q_f \times 10^{-6}$ (F cm$^{-2}$s$_f^n$) | - | $10.88 \pm 3$ | $25.03 \pm 2.2$ | $26.76 \pm 1.8$ |
| $R_p$ ($\Omega$ cm$^2$) | $1466 \pm 60$ | $10{,}423 \pm 300$ | $21{,}806 \pm 700$ | $26{,}822 \pm 900$ |
| $n_f$ | - | $0.559 \pm 0.01$ | $0.557 \pm 0.01$ | $0.63 \pm 0.02$ |
| W ($\Omega^{-1}$ cm$^{-2}$ s$^{0.5}$) | $140.2 \pm 18$ | - | - | - |
| $\eta$ (%) | - | 85.93 | 93.27 | 94.55 |

Inhibition efficiency ($\eta$) can be calculated according to the polarization resistance as shown in Equation (10) [51]:

$$\eta = \frac{R_p - R_p^0}{R_p} \times 100 \tag{10}$$

$R_p^0$ and $R_p$ represent the polarization resistance to corrosion of copper in 3 wt% NaCl solutions in the absence and presence of SCG oil extract, respectively.

The value of $R_p$ serves as an indicator of anti-corrosion ability and can be calculated using Equation (11) [53]:

$$R_p = R_{ct} + R_f \tag{11}$$

The data presented in Table 3 indicate a correlation between the parameters obtained without the inhibitor (blank) and previously published research [24,46,53]. Furthermore, a comparison of the parameters obtained after the addition of the SCG oil extract reveals that the lowest value of charge transfer resistance ($R_{ct}$) (1466 $\Omega$ cm$^2$) was observed for the blank sample, indicating the inhibitory effect of SCG oil on the copper electrode.

Additionally, Table 3 demonstrates that the $R_{ct}$ values increased from $1466 \pm 80$ to $17{,}980 \pm 630$ $\Omega$ cm$^2$ with the rise in SCG oil concentrations. This increase in $R_{ct}$ values with SCG oil concentration can be attributed to a higher surface coverage of the inhibitor, resulting in an improvement in the inhibitor efficiency [54]. In addition, the ($R_f$) value rose from $3698 \pm 7$ $\Omega$ cm$^2$ in the 0.2 g/L SCG oil concentration to $8932 \pm 37$ $\Omega$ cm$^2$ in the 0.6 g/L oil concentration. Consequently, $R_p$ value for the 0.6 g/L SCG oil concentration reached $26{,}822 \pm 900$ $\Omega$ cm$^2$, which is 18-fold higher than that of the blank solution. These findings are consistent with the inhibition efficiency ($\eta$) values presented in Table 3, where the $\eta$ values increased from 85.93% to 94.55% as the oil concentration rose from 0.2 g/L to 0.6 g/L. These results align with the $\eta$ values derived from the potentiodynamic polarization curves recorded around the OCP (Table 2). Hence, it has been demonstrated that the inhibitor forms a stable layer on the copper surface, serving as a proficient impediment, effectively inhibiting corrosion.

### 3.3.4. Adsorption Isotherm Modeling

The inhibition of corrosion by organic molecules is commonly imputed to the adsorption at the metal/solution interface. Adsorption isotherms can provide valuable insights into the interaction between inhibitors and the metal surface. The adsorption of an organic adsorbate at the metal–solution interface can be explained by the substitution of water molecules previously adsorbed on the metallic surface ($H_2O_{(ads)}$) with the organic species in the aqueous solution ($Org_{(sol)}$), as described by the following Equation (12) [55]:

$$Org_{(sol)} + xH_2O_{(ads)} \rightarrow Org_{(ads)} + xH_2O_{(sol)} \tag{12}$$

where $Org_{(sol)}$ and $Org_{(ads)}$ represent the organic compounds present in the aqueous solution and the organic molecules adsorbed onto the metal surface, and x is the ratio indicating the number of water molecules that are substituted by one organic adsorbate molecule [56].

To investigate the adsorption behavior of SCG oil extract on the copper surface, several adsorption isotherms were employed, including Langmuir, Frumkin, Temkin, Freundlich, and Flory Huggins isotherms. To determine the most suitable adsorption isotherm for the experimental data, the correlation coefficient ($R^2$) was calculated. Among them, the Langmuir adsorption isotherm yielded the highest $R^2$ value of 0.999. This indicates that the species present in the SCG oil extract were adsorbed onto the copper surface following the Langmuir adsorption isotherm, according to Equation (13) [25]:

$$\frac{C}{\theta} = \frac{1}{K_{ads}} + C \tag{13}$$

In the adsorption process, the surface coverage ($\theta$) can be determined using the relation $\theta = \eta/100$, where $\eta$ represents the inhibition efficiency expressed as a percentage from the impedance data. The concentration of SCG oil extract in the solution is denoted by C. Furthermore, in the endeavor to ascertain the nature of adsorption, two additional concentrations (0.3 g/L and 0.5 g/L) were subjected to scrutiny.

The adsorption–desorption equilibrium constant is represented by $K_{ads}$, indicating the strength of the adsorption process [25,57].

The Langmuir adsorption isotherm involves the formation of a protective monolayer with a fixed number of active sites on the metal surface [58].

Through plotting $C/\theta$ versus C, a straight line was obtained, as shown in Figure 10. The intercept of this line in the Langmuir adsorption isotherm was used to calculate the adsorption–desorption equilibrium constant, $K_{ads}$, for the SCG oil extract. The significantly high value of $K_{ads}$ (30.959 $g^{-1} \cdot L$) demonstrates the strong adsorption ability of the working oil on the metal surface [59].

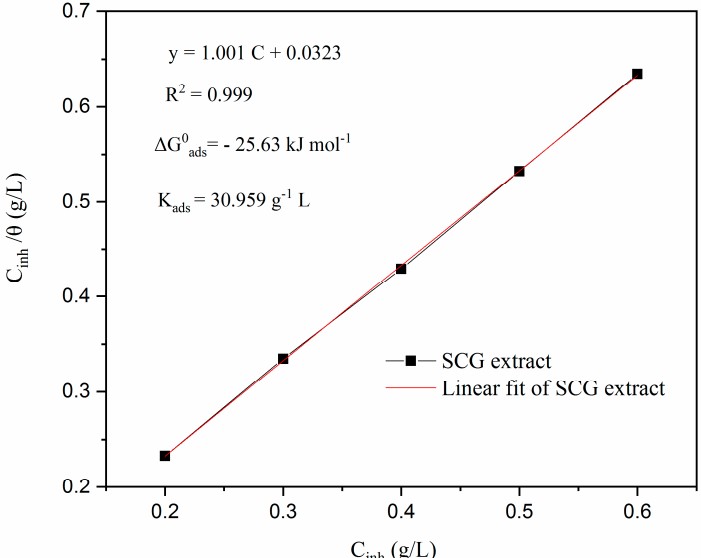

**Figure 10.** Langmuir adsorption isotherm plot and corresponding modeling parameters for SCG oil extract on the copper surface in a 3 wt % NaCl solution at RT.

The adsorption's standard free energy ($\Delta G^0_{ads}$) was calculated from this isotherm through the following Equation [60]:

$$\Delta G^0_{ads} = RT \ln(1000\, K_{ads}) \tag{14}$$

In the above relation, R represents the universal gas constant, T denotes the thermodynamic temperature, and 1000 represents the concentration of water ($g \cdot L^{-1}$). The negative values of $\Delta G^0_{ads}$ ($-25.63$ kJmol$^{-1}$) indicate the spontaneous adsorption process and the strong interaction between the adsorption layer and the copper surface [61].

Typically, the standard free energy of adsorption magnitudes approximately equal to or exceeding $-20$ kJ mol$^{-1}$ could potentially unveil the existence of electrostatic interactions between the inhibitor and the metal surface, thereby signifying a mode of physical adsorption. On the other hand, values around $-40$ kJ mol$^{-1}$ or less indicate coordination between lone pairs of oxygen atoms or $\pi$-electron clouds and the metallic surface, indicating chemical adsorption [56,61].

In the case of our study, the calculated value of $\Delta G^0_{ads}$ falls within the range of $-40$ kJ mol$^{-1}$ to $-20$ kJ mol$^{-1}$, suggesting that the adsorption of the inhibitor on the copper surface occurs through a mixed process involving both physisorption and chemisorption mechanisms [25,62].

## 4. Conclusions

In conclusion, the innovative dimension of our research resides in the transition towards the utilization of natural corrosion inhibitors, as opposed to their synthetic counterparts, through the valorization of biomass waste. The results of this work provide valuable insights into the potential application of SCG extract as an effective and environmentally friendly corrosion inhibitor for copper in saline environments. Through utilizing this bio-oily extract, we not only mitigate the corrosion of copper but also contribute to the sustainable utilization of SCGs, which is currently a major environmental concern due to their improper disposal.

Through the characterization of SCG extract using techniques such as FTIR and TGA, it was determined that the extract contained an oily fraction. The effectiveness of the SCG oil extract as an effective green inhibitor was confirmed through potentiodynamic polarization analysis, which revealed its cathodic-type inhibition mechanism. The organic compounds present in the extract were found to obstruct the cathodic surface sites of copper, leading to inhibition of corrosion. Moreover, the inhibition efficiency of the SCG oil extract rose with higher concentrations of the extract, reaching 96% at a concentration of 0.6 g/L. This observation is consistent with the results obtained from EIS. Furthermore, an evaluation of the adsorption process proved that the adsorption of working oil onto the copper surface involved both physical and chemical interactions, conforming to a Langmuir-type adsorption isotherm.

**Author Contributions:** Methodology, G.R., M.A., A.M. and M.M.; Software, M.M.; Investigation, A.M.; Supervision, G.R. and A.M. All authors have read and agreed to the published version of the manuscript.

**Funding:** This research received no external funding.

**Institutional Review Board Statement:** Not applicable.

**Informed Consent Statement:** Not applicable.

**Data Availability Statement:** Not applicable.

**Acknowledgments:** The authors would like to thank all the members of the laboratory of Electrochemistry and Environment at Sfax National Engineering School and the Research Laboratory of Environmental Sciences and Technologies at the Higher Institute of Environmental Sciences and Technology of Borj Cedria, for supporting this work. Also the authors are grateful to the English professor for his assistance.

**Conflicts of Interest:** The authors declare no conflict of interest.

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
