# Peer review of "Oil Extracted from Spent Coffee Grounds as a Green Corrosion Inhibitor for Copper in a 3 wt% NaCl Solution"

_coatings, doi:10.3390/coatings13101745_

Round 1

Reviewer 1 Report (Previous Reviewer 1)

'

The authors prepared a bio-oil extract from spent coffee grounds as a corrosion inhibitor for copper in a brine solution.  Besides, physiochemical and electrochemical characterization were also conducted to investigate the anticorrosive mechanism.
The subject is interesting and the prepared inhibitors showed a good inhibition effect. The manuscript can be published in its current form.

Author Response

Dear reviewer

Thank you for your comments concerning our manuscript entitled "Oil extracted from spent coffee grounds as a green corrosion inhibitor for copper in a 3 wt% NaCl solution".

Reviewer 2 Report (New Reviewer)

The effect of oil extracted from spent coffee grounds as a green corrosion inhibitor for copper in a 3 wt.% NaCl solution was investigated. The article needs some revision before it can be considered for publication.

Detailed comments:

1. The “Abstract” section needs some revision, which makes it more attractive.

2. The “Introduction section” needs more content. The research done by Velazquez-Torres et al., in particular, should be recounted in more detail. What are the similarities and differences between it and this article, and what are the advantages and disadvantages of each in terms of performance? And whether there are other studies like this.

3. Why “Extensive research in the literature highlights the considerable potential of used coffee grounds in producing value-added energy and non-energy-related products due to their significant content of fatty acids, cellulose, esters, lignin, and hemicelluloses”?

4. What are the reasons for choosing some experimental parameters (a specific mass of the obtained oil was dissolved in tween 80 (m, 1/2 m),3 wt.% NaCl solution, etc.)?

5. Figure 1 needs to be modified to make it more aesthetically pleasing.

6. Figure 5 is not precise and needs to be modified.

7. Why does SCG need pretreatment?

8. From the study, it can be seen that the effect of SCG oil at a concentration of 0.6g/L is the best, so why not continue to increase the concentration to get the best effect and discuss this?

9. The “Conclusion” section lacks comparison, whether there is a significant advantage over other studies. If so, mention it explicitly. And what are the innovative points of this paper also need to be explained.

The English is fine. Some part need corrections.

Author Response

Dear reviewer

Thank you for your comments concerning our manuscript entitled "Oil extracted from spent coffee grounds as a green corrosion inhibitor for copper in a 3 wt% NaCl solution".

Reviewer 3 Report (New Reviewer)

Dear authors, the paper contains valuable data, but the language and style must be improved. Also, some point should be clarified.

Abstract

poses = posses

decoction?

reaching a remarkable 95.78 %

Reviewer: it should be “a 96%” that is not remarkable.

The findings obtained from electrochemical impedance spectroscopy (EIS) were consistent with the inhibitory efficiency order determined by potentiodynamic polarization.

R: Please reformulate

and complies with the Langmuir isotherm.

R: and follows the Langmuir isotherm.

Keywords: Corrosion; copper; spent coffee grounds; voltammetry around OCP; EIS

R: Keywords should be improved, words from the title should not be in a keywords due to better searchability.

1. Introduction

Copper, classified as a noble metal, is widely favored for its remarkable corrosion resistance properties.

Reviewer will not agree. Cu is not a noble metal, and it does not have “remarkable corrosion resistance properties”.

rows 31-40. Whole section should be rewritten.

Taking the corrosion issues into consideration is steadily gaining prominence.

This is not for the scientific paper, this is well known fact.

row 42: The investigation of copper corrosion in aqueous media primarily aimed at serving industrial corrosion control measures [4.org/users/locompounds containing atoms of sulfur, phosphorus, nitrogen, and oxygen constitute a significant proportion of the inhibitors employed.

R: What is the meaning of this sentence?

row 81: To prepare the inhibitor solution, a specific mass of the obtained oil was dissolved in tween 80 (m, 1/2 m),

R: What is the meaning of this sentence?

row 84: Subsequently, the corrosion solution was formulated by incorporating the SCG extract concentrations as inhibitors into a 3 wt% NaCl solution. The investigation included studying the influence of SCG extract concentration and the immersion time of the inhibitor in the chloride solution, all conducted at room temperature (25 ± 2 °C), as depicted in Figure 1 (step 2).

R: Very incomprehensible, and should be given with more details.

The selection of the 90-minute immersion period was based on a preliminary investigation involving five different immersion times (30, 60, 90, 120, and 150 minutes) to provide optimal stability and consistent results.

R: Please provide diagram of the Ecorr over time for bare copper and different concentrations of SCG extract

row 136: Potentiodynamic polarization curves were constructed by plotting the potential from - 400 to 400 mV/SCE at a polarization rate of 0.5 mv / second.

R: ???????

-The mathematical modeling of the potentiodynamic polarization curve (ΔE = ± 60 mV) was performed using Equation (1) …..

R: Why you use this procedure, it is much easier to use polarization resistance +/->20 mV vs Ecorr and formula, that is latter given as Eqs 7 and 8 in R&D section

jcorr= (ba*bc)/(2.3*(ba+bc)*Rp)

Also from Figure 4. it is obvious that bc tend to infinity diffusion controlled oxygen reduction reaction, so Eq is

jcorr= ba/(2.3*ba*Rp)

Figure 2. FTIR spectrums of SCG samples and SCG oil.

Reviewer: There is fourth SSG samples, nothing is mentioned in the experimental how they are prepared.

Lastly, region III, occurring at potentials below -380 mV, can be attributed to the hydrogen evolution reaction as depicted in Equation (3)

R: This is thermodynamically impossible. Reversible potential for hydrogen evolution in neutral media is – 665 mV vs SCE?

3.3.3. Electrochemical impedances spectroscopy (EIS)

In all impedance curves, fitted lines must be added.

Figure 9. Langmuir adsorption isotherm plot….

Three points are insufficient for determining the type of adsorption in a such narrow concentration range.

It is indicated in Comments and Suggestions for Authors

Author Response

Dear reviewer

Thank you for your comments concerning our manuscript entitled "Oil extracted from spent coffee grounds as a green corrosion inhibitor for copper in a 3 wt% NaCl solution".

Reviewer 4 Report (New Reviewer)

The manuscript describes SCG extract as a good corrosion inhibitor for copper in a saline solution. Several experimental characterizations were conducted in the manuscript; however, some parts of the work should be amended to make the discussion of the data sound. The authors should address the following concerns.

1.     The chemical formula of the main components of the SCG extract should be displayed.

2.     How about the dispersion of the SCG extract in the solution?

3.     When the concentration is 0.6 g/L, the corrosion inhibition efficiency is the highest. Why don't you choose a higher concentration?

4.     The surface morphology and chemical composition of the copper after immersion with or without corrosion inhibitor should be analyzed.

5.     The EIS spectra after different immersion time should be tested to evaluate the corrosion resistance of the corrosion inhibitor.

6.     The authors should carefully check the format and spelling, for example, the abscissa title of Figure 4 should be ‘Potential (V vs. SCE)’ rather than ‘Potentiel (V/SCE)’.

The author should carefully check grammar and spelling.

Author Response

Dear reviewer

Thank you for your comments concerning our manuscript entitled "Oil extracted from spent coffee grounds as a green corrosion inhibitor for copper in a 3 wt% NaCl solution".

Round 2

Reviewer 3 Report (New Reviewer)

Dear Authors,

Many thanks for a correct response, and clarification.

Best wishes in further work.

Reviewer 4 Report (New Reviewer)

The authors have addressed most of my concerns. So, I am pleased to recommend this manuscript for acception. 

This manuscript is a resubmission of an earlier submission. The following is a list of the peer review reports and author responses from that submission.

Round 1

Reviewer 1 Report

The authors prepared a bio-oil extract from spent coffee grounds as a corrosion inhibitor for copper in a brine solution.  Besides, physiochemical and electrochemical characterization were also conducted to investigate the anticorrosive mechanism.
Although the subject is interesting and the prepared inhibitors showed a good inhibition effect, there are some problems with this paper that should be addressed before it is considered for publication. They are outlined below:

1.      Fig 3. Increase the size of the number on both axes.

2.      You wrote equation 2 twice.

3.      Fig 5 The Nyquist plot for the blank solution must be highlighted. As it stands, the Warburg-type diffusion process can’t be seen.

4.      The EIS plots should be plotted with the fitting and reporting χ2

5.      Fig 5. Both axes must have the same scale.

6.      The standard deviation was only reported in Table 2 and not in Table 3. Why?

7.      You don’t need to write all the isotherms used. Just report the one that gave you the best fit.

8.      The temperature was not taken into account. No surface morphology results were presented. The metal surface and the product analysis should be conducted including SEM-EDS XRD, and/or XPS, etc.

9.      These results lack detailed mechanism explanations. Hence, the discussion and conclusion must be improved also.

You synthesized a green corrosion inhibitor using as solvent n-hexane. It does not exactly comply with the new strict environmental policies.

Reviewer 2 Report

The authors of this article study influence of the concentration of spent coffee grounds oil extract on the corrosion behaviors of copper in 3 wt% NaCl solution. There are still some problems in the article, and the author is recommended to improve it. It is suggested that this manuscript can be accepted after minor revisions. Listed below are detailed comments on the author's manuscript.

1.       It is proposed to revise the elevated Abstract section.

2.       Page 3 – line 118, Page 7 – line 236: "Where…, respectively." and "Where…, respectively." There are no indented characters at the beginning of the sentence, and it is recommended that they be changed.

3.       It is recommended that the peaks are labelled in Figure 2.

4.       Page 7 – line 239: It is recommended that the presentation of Table 2 and Figure 5 in the text should not be a separate paragraph.

5.       The text in Figure 3 is too small to read and it is recommended that it be amended.

6.     The English language needs revision as there are lots of mistakes.